# Lazy vs hasty: linearization in deep networks impacts learning schedule based on example difficulty

**Thomas George**  *georgeth@mila.quebec*
*Mila, Université de Montréal*

**Guillaume Lajoie**  *g.lajoie@umontreal.ca*
*Mila, Université de Montréal*
*Canada CIFAR AI Chair*

**Aristide Baratin**  *a.baratin@samsung.com*
*Samsung, SAIT AI Lab, Montréal*

**Reviewed on OpenReview:** *https://openreview.net/forum?id=lukVf4VrfP*

## Abstract

Among attempts at giving a theoretical account of the success of deep neural networks, a recent line of work has identified a so-called 'lazy' training regime in which the network can be well approximated by its linearization around initialization. Here we investigate the comparative effect of the lazy (linear) and feature learning (non-linear) regimes on subgroups of examples based on their difficulty. Specifically, we show that easier examples are given more weight in feature learning mode, resulting in faster training compared to more difficult ones. In other words, the non-linear dynamics tends to sequentialize the learning of examples of increasing difficulty. We illustrate this phenomenon across different ways to quantify example difficulty, including c-score, label noise, and in the presence of easy-to-learn spurious correlations. Our results reveal a new understanding of how deep networks prioritize resources across example difficulty.

## 1 Introduction

Understanding the performance of deep learning algorithms has been the subject of intense research efforts in the past few years, driven in part by observed phenomena that seem to defy conventional statistical wisdom (Neyshabur et al., 2015; Zhang et al., 2017; Belkin et al., 2019). Notably, many such phenomena have been analyzed rigorously in simpler contexts of high dimensional linear or random feature models (e.g., Hastie et al., 2022; Bartlett et al., 2021), which shed a new light on the crucial role of overparametrization in the performance of such systems. These results also apply to the so-called *lazy training* regime (Chizat et al., 2019), in which a deep network can be well approximated by its linearization at initialization, characterized by the neural tangent kernel (NTK) (Jacot et al., 2018; Du et al., 2019b; Allen-Zhu et al., 2019). In this regime, deep networks inherit the inductive bias and generalization properties of kernelized linear models.

It is also clear, however, that deep models cannot be understood solely through their kernel approximation. Trained outside the lazy regime (e.g., Woodworth et al., 2020), they are able to learn adaptive representations, with a time-varying tangent kernel (Fort et al., 2020) that specializes to the task during training (Kopitkov & Indelman, 2020; Baratin et al., 2021; Paccolat et al., 2021; Ortiz-Jiménez et al., 2021). Several results showed examples where their inductive bias cannot be characterized in terms of a kernel norm (Savarese et al., 2019; Williams et al., 2019), or where they provably outperform any linear method (Malach et al., 2021). Yet, the specific mechanisms by which the two regimes differ, which could explain the performance gaps often observed in practice (Chizat et al., 2019; Geiger et al., 2020), are only partially understood.

Our work contributes new qualitative insights into this problem, by **investigating the comparative effect of the lazy and feature learning regimes on the training dynamics of various groups of examples of increasing difficulty**. We do so by means of a control parameter that modulates linearity of the parametrization and smoothly interpolates the two training regimes (Chizat et al., 2019; Woodworth et al., 2020). We provide empirical evidence and theoretical insights suggesting that **the feature learning regime puts higher weight on easy examples** at the beginning of training, which results in an increased learning speed compared to more difficult examples. This can be understood as an instance of the **simplicity bias** brought forward in recent work (Arpit et al., 2017; Rahaman et al., 2019; Kalimeris et al., 2019), which we show here to be much more pronounced in the feature learning regime. It also resonates with the old idea that generalization can benefit from some curriculum learning strategy (Elman, 1993; Bengio et al., 2009).

**Contributions**

- We introduce and test the hypothesis of qualitatively different example importance between linear and non-linear regimes;

- Using adequately normalized plots, we present a unified picture using 4 different ways to quantify example difficulty, where easy examples are prioritized in the non-linear regime. We illustrate empirically this phenomenon across different ways to quantify example difficulty, including c-score (Jiang et al., 2021), label noise, and spurious correlation to some easy-to-learn set of features.

- We illustrate some of our insights in a simple quadratic model amenable to analytical treatment.

The general setup of our experiments is introduced in Section 2. Section 3 is our empirical study, which begins with an illustrative example on a toy dataset (Section 3.1), followed by experiments on CIFAR 10 in two setups where example difficulty is quantified using respectively c-scores and label noise (Section 3.2). We also examine standard setups where easy examples are those with strong correlations between their labels and some spurious features (Section 3.3). Section 4 illustrates our findings with a theoretical analysis of a specific class of quadratic models, whose training dynamics is solvable in both regimes. We conclude in Section 5.

**Related work**   The neural tangent kernel was initially introduced in the context of infinitely wide networks, for a specific parametrization that provably leads to the lazy regime (Jacot et al., 2018). Such a regime allows to cast deep learning as a linear model using a fixed kernel, enabling import of well-known results from linear models, such as guarantees of convergence to a global optimum (Du et al., 2019b;a; Allen-Zhu et al., 2019). On the other hand, it is also clear that the kernel regime does not fully capture the behavior of deep models – including, in fact, infinitely wide networks (Yang & Hu, 2021). For example, in the so-called mean field limit, training two-layer networks by gradient descent learns adaptive representations (Chizat & Bach, 2018; Mei et al., 2018) and it can be shown that the inductive bias cannot be characterized in terms of a RKHS norm (Savarese et al., 2019; Williams et al., 2019). Performance gaps between the two regimes are also often observed in practice (Chizat et al., 2019; Arora et al., 2019; Geiger et al., 2020).

Subsequent work showed and analyzed how, for a fixed (finite-width) network, the scaling of the model at initialization also controls the transition between the lazy regime, governed by the empirical neural tangent kernel, and the standard feature learning regime (Chizat et al., 2019; Woodworth et al., 2020; Agarwala et al., 2020). In line with this prior work, in our experiments below we use a scaling parameter $\alpha > 0$ that modulates linearity of the parametrization and allows us to smoothly interpolates between the vanilla training ($\alpha = 1$) where features are learned and the lazy ($\alpha \to \infty$) regime where they are not. We compare training runs with various values of $\alpha$ and empirically assess linearity with several metrics described in Section 2 below.

Our results are in line with a group of work showing how deep networks learn patterns and functions of increasing complexity during training (Arpit et al., 2017; Kalimeris et al., 2019). An instance of this is the so-called spectral bias empirically observed in Rahaman et al. (2019) where a sum of sinusoidal signals is incrementally learned from low frequencies to high frequencies, or in Zhang et al. (2021) that use Fourier decomposition to analyze the function learned by a deep convolutional network on a vision task. The spectral bias is well understood in linear regression, where the gradient dynamics favours the large singular directions of the feature matrix. For neural networks in the lazy regime, spectral analysis of the neural tangent kernel

have been investigated for architectures and data (e.g, uniform data on the sphere) allowing for explicit computations (e.g., decomposition of the NTK in terms of spherical harmonics) (Bietti & Mairal, 2019; Basri et al., 2019; Yang & Salman, 2019). This is a setup where the spectral bias can be rigorously analyzed, i.e., we can get explicit information about the type of functions that are learned quickly and generalize well. In this context, our work specifically focuses on comparing the lazy and the standard feature learning regimes.

Our theoretical model in Section 4 reproduces some of the key technical ingredients of known analytical results (Saxe et al., 2014; Gidel et al., 2019) on deep linear networks. These results show how, in the context of multiclass classification or matrix factorization, the principal components of the input-output correlation matrix are learned sequentially from the highest to the lowest mode. Note however that in the framework of these prior works, the number of modes is bounded by the output dimension - which thus reduces to one in the context of regression or binary classification. By contrast, our theoretical analysis applies to the components of the vector of labels $\boldsymbol{Y} \in \mathbb{R}^n$ in the eigenbasis of the kernel $\boldsymbol{X}\boldsymbol{X}^\top$ defined by the input matrix $\boldsymbol{X} \in \mathbb{R}^{n \times d}$. Thus, despite the technical similarities, our framework approaches the old problem of the relative learning speed of different modes in factorized models from a novel angle. In particular, it allows us to frame the notion of example difficulty in this context.

Example difficulty is a loosely defined concept that has been the subject of intense research recently. Ways to quantify example difficulty for a model/algorithm to learn individual examples include e.g. self-influence (Koh & Liang, 2017), example forgetting (Toneva et al., 2019), TracIn (Pruthi et al., 2020), C-scores (Jiang et al., 2021), or prediction depth (Baldock et al., 2021). We believe that a comprehensive theory of generalization in deep learning will require to understand how neural networks articulate learning and memorization to both fit the head (easy examples) and the tail (difficult examples) of the data distribution (Hooker et al., 2020; Feldman & Zhang, 2020; Sagawa et al., 2020a;b).

## 2 Setup

We consider neural networks $f_{\boldsymbol{\theta}}$ parametrized by $\boldsymbol{\theta} \in \mathbb{R}^p$ (i.e. weights and biases for all layers), and trained by minimizing some task-dependent loss function $\ell(\boldsymbol{\theta}) := \sum_{i=1}^n \ell_i(f_{\boldsymbol{\theta}}(\mathbf{x}_i))$ computed on a training dataset $\{\mathbf{x}_1, \cdots, \mathbf{x}_n\}$, using variants of gradient descent,

$$\boldsymbol{\theta}^{(t+1)} = \boldsymbol{\theta}^{(t)} - \eta \nabla_{\boldsymbol{\theta}} \ell(\boldsymbol{\theta}^{(t)}), \tag{1}$$

with some random initialization $\boldsymbol{\theta}^{(0)}$ and a chosen learning rate $\eta > 0$.

**Linearization** A Taylor expansion and the chain rule give the corresponding updates $f^{(t)} := f_{\boldsymbol{\theta}^{(t)}}$ for any network output, at first order in the learning rate,

$$f^{(t+1)}(\mathbf{x}) \simeq f^{(t)}(\mathbf{x}) - \eta \sum_{i=1}^n K^{(t)}(\mathbf{x}, \mathbf{x}_i) \nabla \ell_i, \tag{2}$$

which depend on the time-varying **tangent kernel** $K^{(t)}(\mathbf{x}, \mathbf{x}') := \nabla_{\boldsymbol{\theta}} f^{(t)}(\mathbf{x})^\top \nabla_{\boldsymbol{\theta}} f^{(t)}(\mathbf{x}')$. The *lazy regime* is one where this kernel remains nearly constant throughout training. Training the network in this regime thus corresponds to training the linear predictor defined by

$$\bar{f}_{\boldsymbol{\theta}}(\mathbf{x}) := f_{\boldsymbol{\theta}^{(0)}}(\mathbf{x}) + (\boldsymbol{\theta} - \boldsymbol{\theta}^{(0)})^\top \nabla_{\boldsymbol{\theta}} f_{\boldsymbol{\theta}^{(0)}}(\mathbf{x}). \tag{3}$$

**Modulating linearity** In our experiments, following Chizat et al. (2019), we modulate the level of "non-linearity" during training of a deep network with a scalar parameter $\alpha \geq 1$, by replacing our prediction $f_{\boldsymbol{\theta}}$ by

$$f_{\boldsymbol{\theta}}^{\alpha}(\mathbf{x}) := f_{\boldsymbol{\theta}^{(0)}}(\mathbf{x}) + \alpha (f_{\boldsymbol{\theta}}(\mathbf{x}) - f_{\boldsymbol{\theta}^{(0)}}(\mathbf{x})) \tag{4}$$

and by rescaling the learning rate as $\eta_\alpha = \eta/\alpha^2$. In this setup, gradient descent steps in parameter space are rescaled by $1/\alpha$ while steps in function space (up to first order) are in $O(1)$ in $\alpha$.[1] $\alpha$ can also be viewed as

---

[1]Under some assumptions such as strong convexity of the loss, it was shown (Chizat et al., 2019, Thm 2.4) that as $\alpha \to \infty$, the gradient descent trajectory of $f_{\boldsymbol{\theta}^{(t)}}^{\alpha}$ gets uniformly close to that of the linearization $\bar{f}_{\boldsymbol{\theta}^{(t)}}$.

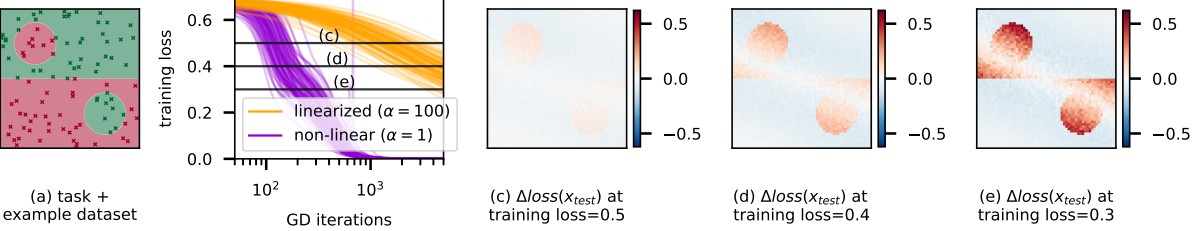

Figure 1: 100 randomly initialized runs of a 4 layers MLP trained on the yin-yang dataset **(a)** using gradient descent in both the non-linear ($\alpha = 1$) and linearized ($\alpha = 100$) setting. The training losses **(b)** show a speed-up in the non-linear regime: in order to compare both regimes at equal progress, we normalize by comparing models extracted at equal training loss thresholds **(c)**, **(d)** and **(e)**. We visualize the differences $\Delta \text{loss} \left(x_{\text{test}}\right) = \text{loss} f_{\text{non-linear}} \left(x_{\text{test}}\right) - \text{loss} f_{\text{linear}} \left(x_{\text{test}}\right)$ for test points paving the 2d square $[-1, 1]^2$ using a color scale. We observe that these differences are not uniformly spread across examples: instead they suggest a comparative bias of the non-linear regime towards correctly classifying easy examples (large areas of the same class), whereas difficult examples (e.g. the small disks) are boosted in the linear regime.

controlling the *level of feature adaptivity*, where large values of $\alpha$ result in linear training where features are not learned. We will also experiment with $\alpha < 1$ below, which enhances adaptivity compared the standard regime ($\alpha = 1$). The goal of this procedure is two-fold: ($i$) to be able to smoothly interpolate between the standard regime at $\alpha = 1$ and the linearized one, ($ii$) to work with models that are nearly linearized yet practically trainable with gradient descent.

**Linearity measures** In order to assess linearity of training runs for empirically chosen values of the re-scaling factor $\alpha$, we track three different metrics during training (fig 2 right):

- **Sign similarity** counts the proportion of ReLUs in all layers that have kept the same activation status (0 or $> 0$) from initialization.

- **Tangent kernel alignment** measures the similarity between the Gram matrix $\boldsymbol{K}_{ij}^{(t)} = K^{(t)}(\mathbf{x}_i, \mathbf{x}_j)$ of the tangent kernel with its initial value $\boldsymbol{K}^{(0)}$, using kernel alignment (Cristianini et al., 2001)

$$\text{KA}(\boldsymbol{K}^{(t)}, \boldsymbol{K}^{(0)}) = \frac{\text{Tr}[\boldsymbol{K}^{(t)}\boldsymbol{K}^{(0)}]}{\|\boldsymbol{K}^{(t)}\|_F \|\boldsymbol{K}^{(0)}\|_F} \quad (\|\cdot\|_F \text{ is the Froebenius norm}) \tag{5}$$

- **Representation alignment** measures the similarity of the last non-softmax layer representation $\phi_R^{(t)}(x)$ with its initial value $\phi_R^{(0)}$, in terms of the kernel alignment (eq. 5) of the corresponding Gram matrices $(\boldsymbol{K}_R^{(t)})_{ij} = \phi_R^{(t)}(\mathbf{x}_i)^\top \phi_R^{(t)}(\mathbf{x}_j)$.

## 3 Empirical Study

### 3.1 A motivating example on a toy dataset

We first explore the effect of modulating the training regime for a binary classification task on a toy dataset with 2d inputs, for which we can get a visual intuition. We use a fully-connected network with 4 layers and ReLU activations. For 100 independent initial parameter values, we generate 100 training examples uniformly on the square $[-1, 1]^2$ from the yin-yang dataset (fig. 1.a). We perform 2 training runs for $\alpha = 1$ and $\alpha = 100$ using (full-batch) gradient descent with learning rate 0.01.

**Global training speed-up and normalization** After the very first few iterations, we observe a speed-up in training progress of the non-linear regime (fig. 1.b). This is consistent with previously reported numerical experiments on the lazy training regime (Chizat et al., 2019, section 3). This raises the question of whether

this acceleration comes from a global scaling in all directions, or if it prioritizes certain particular groups of examples. We address this question by comparing the training dynamics at equal progress: we counteract the difference in training speed by normalizing by the mean training loss, and we compare the linear and non-linear regimes at common thresholds ((c), (d) and (e) horizontal lines in fig. 1.b).

**Comparing linear and non-linear regimes**  At every threshold value, we compute the predictions on test examples uniformly paving the 2d square. We compare both regimes on individual test examples by plotting (fig. 1.c, 1.d, 1.e) the differences in loss values,

$$\Delta \text{loss}\,(x_{\text{test}}) = \text{loss} f_{\text{non-linear}}\,(x_{\text{test}}) - \text{loss} f_{\text{linear}}\,(x_{\text{test}}) \tag{6}$$

Red (resp. blue) areas indicate a lower test loss for the linearized (resp. non-linear) model. Remarkably, the resulting picture is not uniform: these plots suggest that compared to the linear regime, the non-linear training dynamics speeds up for specific groups of examples (the large top-right and bottom-left areas) at the expense of examples in more intricate areas (both the disks and the areas between the disks and the horizontal boundary).

## 3.2   Hastening easy examples

We now experiment with deeper convolutional networks on CIFAR10, in two setups where the training examples are split into groups of varying difficulty. Additional experiments with various other choices of hyperparameters and initialization seed are reported in Appendix F.

### 3.2.1   Example difficulty using C-scores

In this section we quantify example difficulty using consistency scores (C-scores) (Jiang et al., 2021). Informally, C-scores measure how likely an example is to be well-classified by models trained on subsets of the dataset that do not contain it. Intuitively, examples with a high C-score share strong regularities with a large group of examples in the dataset. Formally, given a choice of model $f$, for each (input, label) pair $(x, y)$ in a dataset $\mathcal{D}$, Jiang et al. (2021) defines its empirical consistency profile as:

$$\hat{C}_{\mathcal{D},n}\,(x, y) = \hat{\mathbb{E}}^r_{D \overset{n}{\sim} \mathcal{D} \setminus \{(x,y)\}} \left[ \mathbb{P}\,(f\,(x; D) = y) \right], \tag{7}$$

where $\hat{\mathbb{E}}^r$ is the empirical average over $r$ subsets $D$ of size $n$ uniformly sampled from $\mathcal{D}$ excluding $(x, y)$, and $f(\cdot, D)$ is the model trained on $D$. A scalar C-score is obtained by averaging the consistency profile over various values of $n = 1, \cdots, |\mathcal{D}| - 1$. For CIFAR10 we use pre-computed scores available as https://github.com/pluskid/structural-regularity.

While training, we compute the loss separately on 10 subsets of the training set ranked by increasing C-scores deciles (e.g. examples in the last subset are top-10% C-scores), for both the training set and test set. We also train a linearized copy of this network (so as to share the same initial conditions) with $\alpha = 100$. The $\alpha = 1$ run is trained for 200 epochs (64 000 SGD iterations) whereas in order to converge the $\alpha = 100$ run is trained for 1 000 epochs (320 000 SGD iterations). Similarly to fig. 1, we normalize training progress using the mean training loss in order to compare regimes at equal progress. We check (fig. 2 top right) that the model with $\alpha = 100$ indeed stays in the linear regime during the whole training run, since all 3 linearity metrics that we report essentially remain equal to 1. By contrast, in the non-linear regime ($\alpha = 1$), a steady decrease of linearity metrics as training progresses indicates a rotation of the NTK and the representation kernel, as well as lower sign similarity of the ReLUs.

The results are shown in fig. 2 (top left). As one might expect, examples with high C-scores are learned faster during training than examples with low C-scores in both regimes. Remarkably, this effect is amplified in the non-linear regime compared to the linear one, as we can observe by comparing e.g. the top (resp. bottom) decile in light green (resp dark blue). This illustrates a relative acceleration of the non-linear regime in the direction of easy examples.

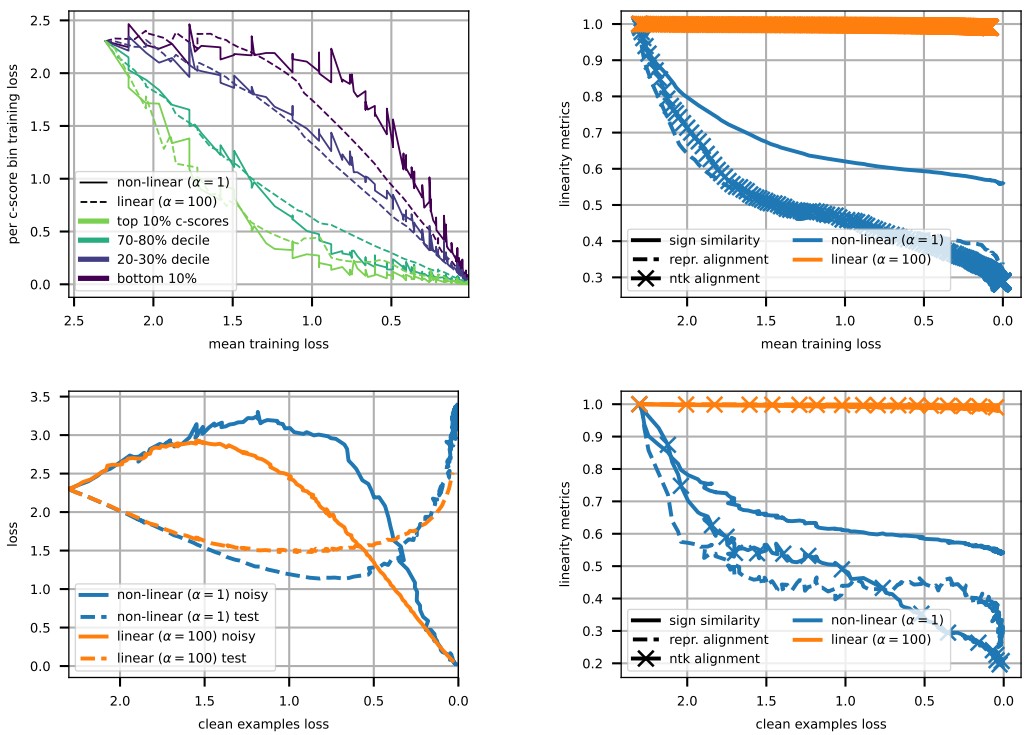

Figure 2: Starting from the same initial parameters, we train 2 ResNet18 models with $\alpha = 1$ (standard training) and $\alpha = 100$ (linearized training) on CIFAR10 using SGD with momentum. **(Top left)** We compute the training loss separately on 10 subgroups of examples ranked by their C-scores. Training progress is normalized by the mean training loss on the $x$-axis. Unsurprisingly, in both regimes examples with high C-scores are learned faster. Remarkably, this ranking is more pronounced in the non-linear regime as can be observed by comparing dashed and solid lines of the same color. **(Bottom left)** We randomly flip the class of 15% of the training examples. At equal progress (measured by equal clean examples loss), the non-linear regime prioritizes learning clean examples and nearly ignores noisy examples compared to the linear regime since the solid curve remains higher for the non-linear regime. Concomitantly, the non-linear test loss reaches a lower value. **(Right)** On the same training run, as a sanity check we observe that the $\alpha = 100$ training run remains in the linear regime throughout since all metrics stay close to 1, whereas in the $\alpha = 1$ run, the NTK and representation kernel rotate, and a large part of ReLU signs are flipped. These experiments are completed in Appendix F with accuracy plots for the same experiments, and with other experiments with varying initial model parameters and mini-batch order.

### 3.2.2 Example difficulty using label noise

In this section we use label noise to define difficult examples. We train a ResNet18 on CIFAR10 where 15% of the training examples are assigned a wrong (random) label. We compute the loss and accuracy independently on the regular examples with their true label, and the noisy examples whose label is flipped. In parallel, we train a copy of the initial model in the linearized regime with $\alpha = 100$. Fig. 2 bottom right shows the 3 linearity metrics during training in both regimes.

The results are shown in fig. 2 (bottom left). In both regimes, the training process begins with a phase where only examples with true labels are learned, causing the loss on examples with wrong labels to increase. A second phase starts (in this run) at 1.25 clean examples loss for the non-linear regime and 1.5 clean loss for the linear regime, where random labels are getting memorized. We see that the first phase takes a larger part of the training run in the non-linear regime than in the linear regime. We interpret this as the fact that the non-linear regime prioritizes learning easy examples. As a consequence, the majority of the training process is dedicated to learning the clean examples in the non-linear regime, whereas in the linear regime both the clean and the noisy labels are learned simultaneously passed the 1.5 clean loss mark. Concomitantly,

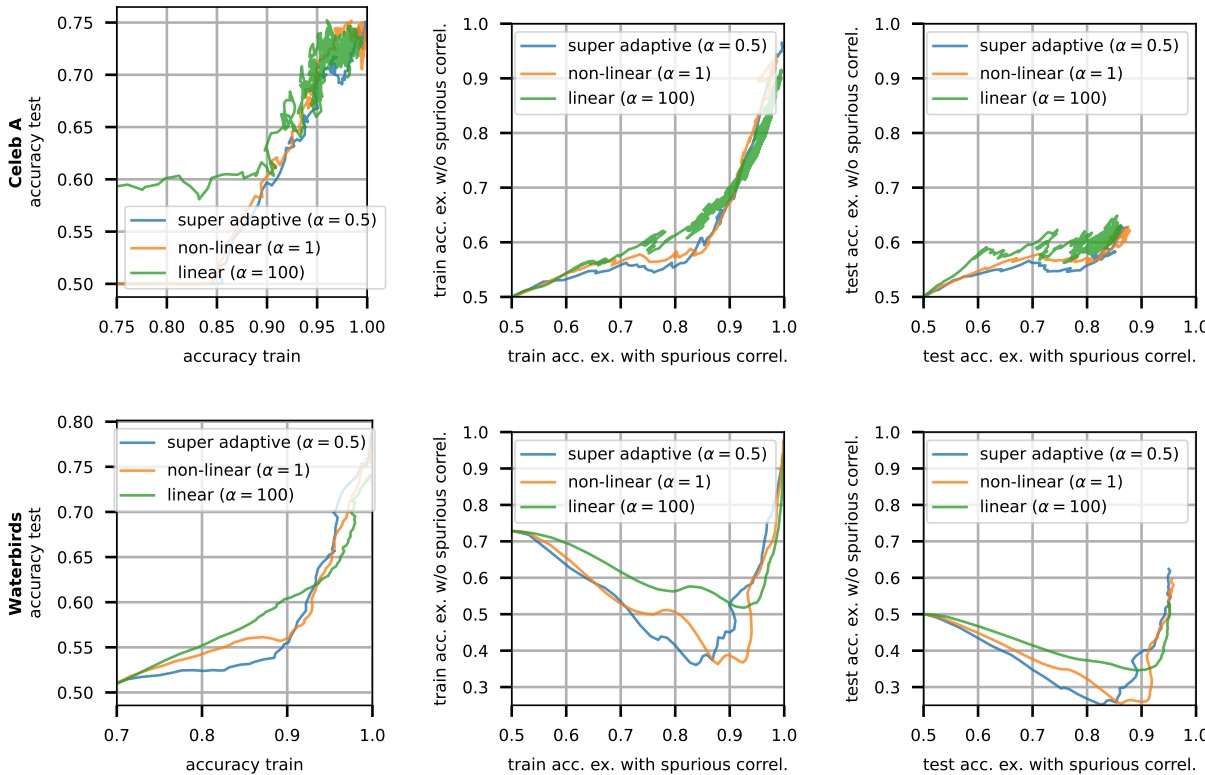

Figure 3: We visualize the trajectories of training runs on 2 spurious correlations setups, by computing the accuracy on 2 separate subsets: one with examples that contain the spurious feature (`with spurious`), the other one without spurious correlations (`w/o spurious`). On Celeb A **(top row)**, the attribute 'blond' is spuriously correlated with the gender 'woman'. In the first phase of training we observe that **(left)** the test accuracy is essentially higher for the linear run, which can be further explained by observing that **(middle)** the training accuracy for `w/o spurious` examples increases faster in the linear regime than in non-linear regimes at equal `with spurious` training accuracy. **(right)** A similar trend holds for test examples. In this first part the linear regime is less sensitive to the spurious correlation (easy examples) thus gets better robustness. **(bottom row)** On Waterbirds, the background (e.g. a lake) is spuriously correlated with the label (e.g. a water bird). **(left)** We observe the same hierarchy between the linear run and other runs. In the first training phase, the linear regime is less prone to learning the spurious correlation: the `w/o spurious` accuracy stays higher while the `with spurious` examples are learned (**(middle)** and **(right)**). These experiments are completed in fig. 12 in Appendix F with varying initial model parameters and mini-batch order.

comparing the sweet spot (best test loss) of the two regimes indicates a higher robustness to label noise thus a clear advantage for generalization in the non-linear regime.

## 3.3   Spurious correlations

We now examine setups where easy examples are those with strong correlations between their labels and some spurious feature (Sagawa et al., 2020b). We experiment with CelebA (Liu et al., 2015) and Waterbirds (Wah et al., 2011) datasets.

**CelebA**   (Liu et al., 2015) is a collection of photographs of celebrities' faces, each annotated with its attributes, such as hair color or gender. Similarly to Sagawa et al. (2020b), our task is to classify pictures based on whether the person is blond or not. In this dataset, the attribute "the person is a woman" is spuriously correlated with the attribute "the person is blond", since the attribute "blond" is over-represented among women (24% are blond) compared to men (2% are blond).

We use 20 000 examples of CelebA to train a ResNet18 classifier on the task of predicting whether a person is blond or not, using SGD with learning rate 0.01, momentum 0.9 and batch size 100. We also extract a balanced dataset with 180 (i.e. the total number of blond men in the test set) examples in each of 4 categories: blond man, blond woman, non-blond man and non-blond woman. While training progresses, we measure the loss and accuracy on the subgroups man and woman. Starting from the same initial conditions, we train 3 different classifiers with $\alpha \in \{.5, 1, 100\}$.

**Waterbirds**  (Wah et al., 2011) is a smaller dataset of pictures of birds. We reproduce the experiments of Sagawa et al. (2020a) where the task is to distinguish land birds and water birds. In this dataset, the background is spuriously correlated with the type of bird: water birds are typically photographed on a background such as a lake, and similarly there is generally no water in the background of land birds. There are exceptions: a small part of the dataset consists of land birds on a water background and vice versa (e.g. a duck walking in the grass).

We use 4 795 training examples of the Waterbirds dataset. Since the dataset is smaller, we start from a pre-trained ResNet18 classifier from default PyTorch models (pre-trained on ImageNet). We replace the last layer with a freshly initialized binary classification layer, and we set batch norm layers to evaluation mode[2] We train using SGD with learning rate 0.001, momentum 0.9 and minibatch size 100.

From the training set, we extract a balanced dataset with 180 examples in each of 4 groups: land birds on land background, land birds on water background, water bird on land background, and water bird on land background, which we group in two sets: `with spurious` when the type of bird and the background agree, and `w/o spurious` otherwise. While training, we measure the accuracy separately on these 2 sets. We train 3 different classifiers with $\alpha \in \{.5, 1, 100\}$.

Looking at fig. 3 left, for both the Waterbirds and the CelebA experiments we identify two phases: in the first phase the test accuracy is higher for the linear $\alpha = 100$ run than for other runs. In the second phase all 3 runs seem to converge, with a slight advantage for non-linear runs in Waterbirds. Taking a closer look at the first phase in fig. 3 middle and right, we understand this difference in test accuracy in light of spurious and non-spurious features: in the non-linear regime, the training dynamics learns the majority examples faster, at the cost of being more prone to spurious correlations. This can be seen both on the balanced training set and the balanced test set.

## 4  Theoretical Insights

The goal here is to illustrate some of our insights in a simple setup amenable to analytical treatment.

### 4.1  A simple quadratic model

We consider a standard linear regression analysis: given $n$ input vectors $\mathbf{x}_i \in \mathbb{R}^d$ with their corresponding labels $y_i \in \mathbb{R}$, $1 \le i \le n$, the goal is to fit a linear function $f_{\boldsymbol{\theta}}(\mathbf{x})$ to the data by minimizing the least-squares loss $\ell(\boldsymbol{\theta}) := \frac{1}{2}\sum_i (f_{\boldsymbol{\theta}}(\mathbf{x}_i) - y_i)^2$. We will focus on a specific quadratic parametrization, which can be viewed as a subclass of two-layer networks with linear activations.

**Notation**  We denote by $\boldsymbol{X} \in \mathbb{R}^{n \times d}$ the matrix of inputs and by $\boldsymbol{y} \in \mathbb{R}^n$ the vector of labels. We consider the singular value decomposition (SVD),

$$\boldsymbol{X} = \boldsymbol{U}\boldsymbol{M}\boldsymbol{V}^\top := \sum_{\lambda=1}^{r_X} \sqrt{\mu_\lambda}\boldsymbol{u}_\lambda \boldsymbol{v}_\lambda^\top \tag{8}$$

where $\boldsymbol{U} \in \mathbb{R}^{n \times n}, \boldsymbol{V} \in \mathbb{R}^{d \times d}$ are orthogonal and $\boldsymbol{M}$ is rectangular diagonal. $r_X$ denotes the rank of $\boldsymbol{X}$, $\mu_1 \ge \cdots \ge \mu_{r_X} > 0$ are the non zero (squared) singular values; we also set $\mu_\lambda = 0$ for $r_X < \lambda \le \max(n, d)$.

---

[2]In order not to interfere with $\alpha$ scaling (see section 2), and since we consider that batch norm is a whole different theoretical challenge by itself, we chose to turn it off, and instead keep the mean and variance buffers to their pre-trained value. See appendix D for further discussion of this point.

Left and right singular vectors extend to orthonormal bases $(\boldsymbol{u}_1, \cdots, \boldsymbol{u}_n)$ and $(\boldsymbol{v}_1, \cdots, \boldsymbol{v}_d)$ of $\mathbb{R}^n$ and $\mathbb{R}^d$, respectively. In what follows we assume, without loss of generality[3], that the vector of labels has positive components in the basis $\boldsymbol{u}_\lambda$, i.e., $y_\lambda := \boldsymbol{u}_\lambda^\top \boldsymbol{y} \geq 0$ for $\lambda = 1, \cdots n$.

**Parametrization** We consider the following class of functions

$$f_{\boldsymbol{\theta}}(\mathbf{x}) = \boldsymbol{\theta}^\top \mathbf{x}, \quad \boldsymbol{\theta} := \frac{1}{2} \sum_{\lambda=1}^{d} w_\lambda^2 \, \boldsymbol{v}_\lambda \tag{9}$$

In this setting, the least-squares loss is minimized by gradient descent over the vector parameter $\mathbf{w} = [w_1, \cdots w_d]^\top$. Given an initialization $\mathbf{w}^0$, we want to compare the solution of the vanilla gradient (non linear) dynamics with the solution of the lazy regime, which corresponds to training the linearized function (3), given in our setting by $\bar{f}_{\bar{\boldsymbol{\theta}}}(\mathbf{x}) = \bar{\boldsymbol{\theta}}^\top \mathbf{x}$ where $\bar{\boldsymbol{\theta}} = \sum_{\lambda=1}^{d} w_\lambda w_\lambda^0 \, \boldsymbol{v}_\lambda$.

## 4.2 Gradient dynamics

For simplicity, we analyze the continuous-time counterpart of gradient descent,

$$\mathbf{w}(0) = \mathbf{w}^0, \quad \dot{\mathbf{w}}(t) = -\nabla_{\mathbf{w}} \ell(\boldsymbol{\theta}(t)) \tag{10}$$

where the dot denotes the time-derivative. Making the gradients explicit and differentiating (9) yield

$$\dot{\boldsymbol{\theta}}(t) = -\boldsymbol{\Sigma}(t)(\boldsymbol{\theta}(t) - \boldsymbol{\theta}^*), \quad \boldsymbol{\Sigma}(t) = \boldsymbol{V} \mathrm{Diag}(\mu_1 w_1^2(t), \cdots, \mu_d w_d^2(t)) \boldsymbol{V}^\top \tag{11}$$

where $\boldsymbol{\theta}^*$ is the solution of the linear dynamics.[4] Note how $\boldsymbol{\Sigma}(t)$ is obtained from the input correlation matrix $\boldsymbol{X}^\top \boldsymbol{X}$ by rescaling each eigenvalue $\mu_\lambda$ by the time-varying factor $w_\lambda^2$. By contrast, the lazy regime is described by the equation obtained from (11) by replacing $\boldsymbol{\Sigma}(t)$ the constant matrix $\boldsymbol{\Sigma}(0)$.

In the proposition below (proved in Appendix A), we consider the system (10, 11) initialized as $\boldsymbol{\theta}^0 := \frac{1}{2} \sum_{\lambda=1}^{d} (w_\lambda^0)^2 \, \boldsymbol{v}_\lambda$ where we assume that $w_\lambda^0 \neq 0$ for all $\lambda$. We denote by $\tilde{y}_\lambda$ the components of the input-label correlation vector $\boldsymbol{X}^\top \boldsymbol{y} \in \mathbb{R}^d$ in the basis $\boldsymbol{v}_\lambda$: we have $\tilde{y}_\lambda = \sqrt{\mu_\lambda} y_\lambda$ for $1 \leq \lambda \leq r_X$ and 0 when $r_X < \lambda \leq d$.

**Proposition 1** *The solution of (10, 11) is given by,*

$$\boldsymbol{\theta}(t) = \sum_{\lambda=1}^{d} \theta_\lambda(t) \boldsymbol{v}_\lambda, \qquad \theta_\lambda(t) = \begin{cases} \dfrac{\theta_\lambda^0 \theta_\lambda^*}{\theta_\lambda^0 - e^{-2\tilde{y}_\lambda t}(\theta_\lambda^0 - \theta_\lambda^*)} & \text{if } \theta_\lambda^* \neq 0 \\[2mm] \dfrac{\theta_\lambda^0}{1 + 2\mu_\lambda \theta_\lambda^0 t} & \text{if } \theta_\lambda^* = 0 \end{cases} \tag{12}$$

*By contrast, the solution in the linearized regime where $\boldsymbol{\Sigma}(t) \approx \boldsymbol{\Sigma}(0)$ is,*

$$\theta_\lambda(t) = \theta_\lambda^* + e^{-\mu_\lambda \theta_\lambda^0 t}(\theta_\lambda^0 - \theta_\lambda^*) \tag{13}$$

## 4.3 Discussion

While the gradient dynamics (12,13) converge to the same solution $\boldsymbol{\theta}^*$, we see that the convergence rates of the various modes differ in the two regimes. To quantify this, for each dynamical mode $1 \leq \lambda \leq r_X$ such that $|\theta_\lambda^*| \neq 0$, let $t_\lambda(\epsilon), t_\lambda^{\mathrm{lin}}(\epsilon)$ be the times required for $\theta_\lambda$ to be $\epsilon$-close to convergence, i.e $|\theta_\lambda - \theta_\lambda^*| = \epsilon$, in the two regimes. Substituting into (12) and (13) we find that, close to convergence $\epsilon \ll \theta_\lambda^*$ and for a small initialization, $\theta_\lambda^0 \ll \theta_\lambda^*$,

$$t_\lambda(\epsilon) = \frac{1}{\tilde{y}_\lambda} \log \frac{\theta_\lambda^*}{\epsilon \theta_\lambda^0}, \qquad t_\lambda^{\mathrm{lin}}(\epsilon) = \frac{1}{\mu_\lambda \theta_\lambda^0} \log \frac{\theta_\lambda^*}{\epsilon} \tag{14}$$

up to terms in $O(\epsilon/\theta_\lambda^*, \theta_\lambda^0/\theta_\lambda^*)$. Two remarks are in order:

---

[3]One can use the reflection invariance $\sqrt{\mu_\lambda} \boldsymbol{u}_\lambda \boldsymbol{v}_\lambda^\top = \sqrt{\mu_\lambda}(-\boldsymbol{u}_\lambda)(-\boldsymbol{v}_\lambda)^\top$ of the SVD to flip the sign of $y_\lambda$.

[4]Explicitly, $\boldsymbol{\theta}^* = \boldsymbol{\Sigma}^+ \boldsymbol{X}^\top \boldsymbol{y} + P_\perp(\boldsymbol{\theta}^0)$ where $\boldsymbol{\Sigma}^+$ is the pseudoinverse of the input correlation matrix $\boldsymbol{\Sigma} = \boldsymbol{X}^\top \boldsymbol{X}$ and $P_\perp$ projects onto the null space of $\boldsymbol{X}$. It decomposes as $\boldsymbol{\theta}^* = \sum_{\lambda=1}^{d} \theta_\lambda^* \boldsymbol{v}_\lambda$ in the basis $\boldsymbol{v}_\lambda$, where $\theta_\lambda^* = y_\lambda/\sqrt{\mu_\lambda}$ for $1 \leq \lambda \leq r_X$ and $\theta_\lambda^* = \theta_\lambda^0$ for $r_X < \lambda \leq d$.

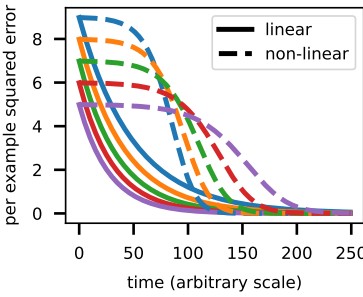 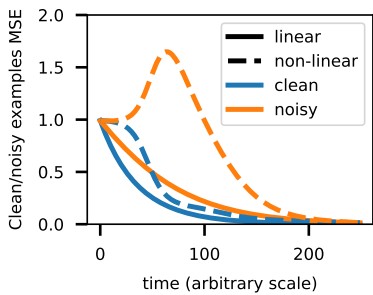 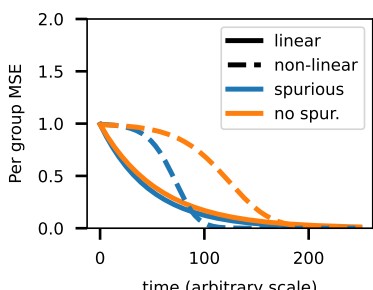

Figure 4: **(left)** Different input/label correlation (example 1): examples are learned in a flipped order in the two regimes. **(middle)** Label noise (example 2): the non-linear dynamics prioritizes learning the clean labels **(right)** Spurious correlations (example 3): the non-linear dynamics prioritizes learning the spuriously correlated feature. These analytical curves are completed with numerical experiments on standard (dense) 2-layer MLP in figure 14 in appendix G, which shows a similar qualitative behaviour.

**Linearization impacts learning schedule.** Specifically, while the learning speed of each mode depends on the components of the label vector in the non linear regime, through $\tilde{y}_\lambda = \sqrt{\mu_\lambda} y_\lambda$, it does not in the linearized one. Thus, the ratio of $\epsilon$-convergence times for two given modes $\lambda, \lambda'$ is $t_\lambda/t_{\lambda'} = \tilde{y}_\lambda/\tilde{y}_{\lambda'}$ in the non-linear case and $t_\lambda^{\mathrm{lin}}/t_\lambda^{\mathrm{lin}} = \mu_{\lambda'}\theta_{\lambda'}^0/\mu_\lambda\theta_\lambda^0$ in the linearized case, up to logarithmic factors.

**Sequentialization of learning.** Non-linearity, together with a vanishingly small initialization, induces a sequentialization of learning for the various modes (see also Gidel et al., 2019, Thm 2). To see this, pick a mode $\lambda$ and consider, for any other mode $\lambda'$, the value $\theta_{\lambda'}(t_\lambda)$ at the time $t_\lambda := t_\lambda(\epsilon)$ where $\lambda$ reaches $\epsilon$-convergence. Let us also write $\theta_\lambda^0 = \sigma\tilde{\theta}_\lambda^0$ where $\sigma$ is small and $\tilde{\theta}_\lambda^0 = O(1)$ as $\sigma \to 0$. Then elementary manipulations show the following:

$$\theta_{\lambda'}(t_\lambda) = \frac{\theta_{\lambda'}^* \tilde{\theta}_{\lambda'}^0}{\tilde{\theta}_{\lambda'}^0 + \left[\epsilon\tilde{\theta}_{\lambda'}^0/\theta_{\lambda'}^*\right]^{2\frac{\tilde{y}_{\lambda'}}{\tilde{y}_\lambda}} \sigma^{2\frac{\tilde{y}_{\lambda'}}{\tilde{y}_\lambda}-1}} =_{\sigma \to 0} \begin{cases} \theta_{\lambda'}^* & \tilde{y}_{\lambda'} > \tilde{y}_\lambda \\ 0 & \tilde{y}_{\lambda'} < \tilde{y}_\lambda \end{cases} \tag{15}$$

In words, for fixed $\epsilon > 0$ and in the limit of small initialization, the mode $\lambda$ gets $\epsilon$-close to convergence before any of the subdominant mode deviates from their (vanishing) initial value: the modes are learned sequentially.

### 4.4 Mode vs example difficulty

We close this section by illustrating the link between *mode* and *example difficulty* on three concrete examples of structure for the training data $\{\mathbf{x}_i, y_i\}_{i=1}^n$. In what follows, we consider the overparametrized setting where $d \geq n$. We denote by $\boldsymbol{e}_1, \cdots \boldsymbol{e}_d$ the canonical basis of $\mathbb{R}^d$.

For each of the examples below, we consider an initialization $\mathbf{w}^0 = \sqrt{\sigma}[1 \cdots 1]^\top$ with small $\sigma > 0$ and the corresponding solutions in Prop. 1 for both regimes.

**Example 1** We begin with a rather trivial setup where each mode corresponds to a training example. It will illustrate how non-linearity can *reverse* the learning order of the examples. Given a sequence of strictly positive numbers $\mu_1 \geq \cdots \mu_n > 0$, we consider the training data,

$$\mathbf{x}_i = \sqrt{\mu_i}\boldsymbol{e}_i, \quad y_i = \mu_{n-i+1}/\sqrt{\mu_i}, \qquad 1 \leq i \leq n \tag{16}$$

In the linearized regime, $f_{\boldsymbol{\theta}(t)}(\mathbf{x}_i)$ converges to $y_i$ at the linear rate $\sigma\mu_i$; the model learns faster the examples with higher $\mu_i$, hence with *lower* index $i$. In the non linear regime, the examples are learned sequentially according to the value $\tilde{y}_i = \sqrt{\mu_i}y_i = \mu_{n-i+1}$, hence from *high to low* index $i$. Thus in this setting, linearization flips the learning order of the training examples (see fig. 4 left).

In examples 2 and 3 below we aim at modelling situations where the labels depend on low-dimensional (in this case, one dimensional) projections on the inputs (e.g. an image classification problem where the labels

mainly depend on the low frequencies of the image). These two examples mirror the two sets of experiments in Sections 3.2.2 and 3.3, respectively.

**Example 2** We consider a simple classification setup on linearly separable data with label noise. Here we assume $d > n$. Conditioned on a set of binary labels $y_i = \pm 1$, the inputs are given by

$$\mathbf{x}_i = \kappa_i y_i \boldsymbol{e}_1 + \eta \boldsymbol{e}_{i+1} \quad 1 \le i \le n \tag{17}$$

where $\kappa_i = \pm 1$ is some 'label flip' variable and $\eta > 0$. We assume we have $q$ 'noisy' examples with flipped labels, i.e. $\kappa_i = -1$, where $1 \le q < \lceil n/2 \rceil$.

The SVD of the feature matrix $\boldsymbol{X} \in \mathbb{R}^{n \times d}$ defined by (17) can be made explicit. In particular, the top left singular vector is $\boldsymbol{u}_1 = \bar{\boldsymbol{y}}/\sqrt{n}$, where $\bar{\boldsymbol{y}} \in \mathbb{R}^n$ denotes the vector of noisy labels $\bar{y}_i := \kappa_i y_i$. This singles out a dominant mode $\boldsymbol{y}_1 := (\boldsymbol{u}_1^\top \boldsymbol{y})\boldsymbol{u}_1$ of the label vector $\boldsymbol{y}$ that is *learned first* by the non-linear dynamics. Explicitly,

$$\boldsymbol{y}_1 = (1 - \frac{2q}{n})\bar{\boldsymbol{y}} \tag{18}$$

For a small noise ratio $q/n \ll 1$, this yields $y_{1i} \approx y_i$ for clean examples and $-y_i$ for noisy ones: fitting the dominant mode amounts to learning the clean examples – while assigning the wrong label to the noisy ones. This is illustrated in fig 4 (middle plot).

**Example 3** We consider a simple spurious correlation setup (Sagawa et al., 2020b), obtained by adding to (17) a 'core' feature that separates all training points. Here we assume $d > n + 1$. Conditioned on a set of binary labels $y_i = \pm 1$, the inputs are given by

$$\mathbf{x}_i = \kappa_i y_i \boldsymbol{e}_1 + \lambda y_i \boldsymbol{e}_2 + \eta \boldsymbol{e}_{i+2}, \quad 1 \le i \le n \tag{19}$$

for some binary variable $\kappa_i = \pm 1$, scaling factor $\lambda \in (0, 1)$, and $\eta > 0$. Given $1 \le q < \lceil n/2 \rceil$, we assume we have a majority group of $n - q$ training examples with $\kappa_i = 1$, whose label is spuriously correlated with the spurious feature $\boldsymbol{e}_2$, and a minority group of $q$ examples with $\kappa_i = -1$.

The analysis is similar as in Example 2. For small noise ratio and scaling factor $\lambda$, the non-linear dynamics enhances the bias towards fitting first the majority group of ('easy') examples with spuriously correlated labels. This illustrates an increased sensitivity of the non linear regime to the spurious feature – at least in the first part of training. This is shown in fig 4 (right plot).

## 5 Conclusion

The recent emphasis on the lazy training regime, where deep networks behave as linear models amenable to analytical treatment, begs the question of the specific mechanisms and implicit biases which differentiates it from the full-fledged feature learning regime of the algorithms used in practice. In this paper, we investigated the comparative effect of the two regimes on subgroups of examples based on their difficulty. We provided experiments in various setups suggesting that easy examples are given more weight in non-linear training mode (deep learning) than in linear training, resulting in a comparatively higher learning speed for these examples. We illustrated this phenomenon across various ways to quantify examples difficulty, through c-scores, label noise, and correlations to some easy-to-learn spurious features. We complemented these empirical observations with a theoretical analysis of a quadratic model whose training dynamics is tractable in both regimes. We believe that our findings makes a step towards a better understanding of the underlying mechanisms that drive the good generalization properties observed in practice.

**Broader Impact Statement**

This work proposes to improve our understanding of the mechanisms behind deep learning models training. There is no clear direct application of these theoretical observations to create harmful tools, however as for any technology, machine learning models can be used for malicious intentions. We also acknowledge that deep learning uses a significant amount of compute capacity when scaled to global tech companies, which in some places is powered by carbon emitting power plants, thus contributes to global warming.

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
