# OpenReview forum: "Lazy vs hasty: linearization in deep networks impacts learning schedule based on example difficulty"
_TMLR — Accepted by TMLR_

### Review · Reviewer_93fy · 2022-10-25

**Summary Of Contributions:**

The authors study the difference between the lazy and feature learning regimes from the perspective of image difficulty. The authors empirically show that the network learns easy samples faster both in the linear and feature regimes; however, the difference between the learning speeds of the easy vs. hard groups is more pronounced in the feature learning regime. The contributions are:

* The authors use three metrics to check whether the network stays in the linear regime despite being finite-width,
* Learning curves are presented in both regimes for CIFAR-10, a small subset of CelebA and Waterbirds,
* A quadratic model is introduced and solved analytically which is then used to study numerically learning speeds in groups for three toy example datasets.

**Audience:**

Yes

**Broader Impact Concerns:**

NA.

**Claims And Evidence:**

Yes

**Requested Changes:**

* Discussion related to Figure 3 needs to be improved. Why are the learning curves in the middle for the Waterbirds dataset non-monotonic compared to CelebA? Also, the training accuracy starts at 0.7 which is very high for the first epoch due to the small dataset size.

* How do the results compare with the gradient starvation paper from NeurIPS-2021 [1]?

*Minor changes:*

* Contributions point 2, the first sentence needs improvement. We present a unified picture, adequately normalized visualizations, etc. are too bombastic. If the toy dataset example in Fig. 1 is referred to here, please explicitly write it so.
* Figure 2 right panel figures are hard to read. Using smaller markers should help.
* Figure 4 middle figure can be plotted for different levels of label corruption $q$.
* In Section 4.4., why only use small $\sigma$ for initialization? A larger $\sigma$ should remove the loss plateaus at the beginning of training in the feature regime, thus the reverse sequentialization phase should be more visible.

[1] Pezeshki, Mohammad, et al. "Gradient starvation: A learning proclivity in neural networks." Advances in Neural Information Processing Systems 34 (2021): 1256-1272.

**Strengths And Weaknesses:**

*Strengths:*

* Using diverse datasets and metrics reduces the bias due to the metric or architecture used to calculate the image difficulty score.
* Solid check with three metrics to make sure that the network stays in the linear regime

*Weaknesses:*

* The major weakness is that the difference between the linear and feature learning regimes is tiny. Even in the linear regime, I understand that sequentialization happens by looking at the learning curves. However from reading the text, it may sound like this only happens in the feature learning regime.
* The title rhymes very well, but --hasty-- sounds like a negative property of feature learning, but I'm not sure if the authors made any such claim.
* The datasets used in spurious correlation experiments are small: overall $180 \times 4 = 720 $ samples. It is not hard to believe that the results would generalize to larger sample sizes given the CIFAR10 results, but the small dataset size weakens the results for spurious correlation experiments.

---

> ### Author Response · Authors · 2022-11-08
> **Thank you for the construcive comments.**
>
> We thank the reviewer for a constructive comment and for their time at reviewing our paper. We hope that our comments below, and our revision of the paper answer your concerns.
>
> *The difference between the linear and feature learning regimes is tiny.*
>
> We respectfully disagree with the reviewer. For instance, this difference makes for a 0.3 difference in the best test loss reached during training in the experiment with noisy labels (figure 2, bottom left). More generally,  we think there is an interest in giving a qualitative difference between the linear regime which has been thoroughly analyzed by recent literature, and the true learning regime for which our understanding of the generalization property still lacks behind.
>
> *Datasets in the spurious correlation experiments are too small*
>
> The small datasets mentioned in section 3.3 are not used for training, but they are used to compute the loss and accuracy on balanced datasets during training. Since there are few examples of the minority group, and we need a balanced dataset, then we could not use more examples. The training datasets are however much bigger: 20,000 examples for CelebA and 4,795 examples for Waterbirds. We clarified this point in the description of the experiments in section 3.3 in the revision.
>
> Regarding learning curves in figure 3:
>
> The learning curves in the Waterbird experiment are non-monotonic, and in particular while training progresses, the "training accuracy w/o spurious" goes below the 50% (random guess) mark. It means that in that first part of training, improving on examples with the spurious feature (i.e. background correlated to type of bird) makes prediction on other examples (background anti-correlated to type of bird) worse. It indicates that this spurious feature is easier to use for the algorithm than in the CelebA experiment, where improving on w/o spurious examples has less consequence on other examples. This probably means that the spurious feature in Waterbids is easier to pick up by the training algorithm.
>
> Regarding comparison with the gradient starvation paper:
>
> The mechanism at play in the GS paper is a side effect of the cross-entropy loss, which is theoretically studied in a (NTK-kernelized) linear model in the paper. We instead study the differences between linear and non-linear regimes. Despite only discussing classification experiments that use the cross-entropy loss in the paper, we also experimented with the MSE loss and observed the same difference in relative learning speed of easy and difficult examples, so we are discussing a different effect, even if the interplay between both is an interesting potential future direction of research.
>
> *A larger initialization  should remove the loss plateaus*
>
> A small initialization is essential to obtain the sequentialization result (15), which occurs in the limit $\sigma →0$. More generally,  as we mention in the related work section, initialization scale is a confounding effect in our analysis, since it is known to also govern the transition between lazy and rich regime (Chizat et al., 2019; Woodworth et al., 2020; Agarwala et al., 2020).

---

### Review · Reviewer_5RX7 · 2022-10-27

**Summary Of Contributions:**

In this work, the authors examine the differences between linearized and non-linear models, with respect to which examples are learned first. In a variety of settings, they present evidence that easier examples are learned first in non-linear models - or perhaps more accurately, easier examples contribute to a larger fraction of the loss. They show this for various definitions of "easiness", including an intuitive picture in the yin-yang model, the  The authors also present a toy model which provably shows similar behavior.

**Audience:**

Yes

**Claims And Evidence:**

Yes

**Requested Changes:**

For Figures 1 and 2, it would also be informative to see plots in terms of classification accuracy. One of the interesting features about classification problems, especially those trained with cross-entropy loss is that the loss is only _correlated_ with the actual objective, the accuracy. It would be good to understand whether or not the effect is driven by non-linear models having more certainty in the easier datapoints, versus the classification being more correct on the easier datapoints. In particular it would be good to look at the accuracies as a function of c-score for the same number of GD steps, for optimal learning rates for the different models. I believe that this is a critical change.

Figure 2, top left, is currently hard to parse. I don't feel confident looking at that plot and coming to the conclusions of the authors. Perhaps it would be useful to also have a plot of the differences between the models? Also, there should be a colorbar indicating the C-score.

For Figure 2, bottom, I didn't quite understand why the gap in the solid curves implied that easier examples were being learned first. Some more clarification on why "increasing loss more on noisy data" means "easier examples are learned first" would be quite helpful.

Figure 3 shows some curves with $\alpha <1$ - a setup which was described and analyzed in https://arxiv.org/abs/2010.07344, which should be added to the references. It would be interesting to included curves with other alpha values in Figures 1 and 2 (more specifically 2, I think figure 1 in its current form is simple and effective). This is especially interesting because, by the above reference, for $\alpha <1$ the optimal learning rate goes as $1/\alpha$ ($1/\beta$ in the paper), which means that the change in loss slows down (in terms of GD steps). In this work my understanding is that the $\alpha = 1$ setting always trains faster than the $\alpha = 100$ setting.

It would also be useful to have an experiment on a dataset-model pair where the non-linear model performs significantly better than the linearized model after training - or at least shows accuracy differences according to example difficulty. The results of Figure 3 feel a bit weak because the curves diverge and then converge again, as it appears that the linearized model and the non-linear model learn to similar accuracies. Perhaps creating similar accuracy curves in the CIFAR problem, stratified by c-scores, would be sufficient. I think this is a critical change.

**Strengths And Weaknesses:**

The basic concept of the paper, and the subsequent experiments are very sound. In particular, the authors did a good job of considering multiple ways to define difficulty of examples, and attempted to properly normalize comparisons by, for example, grouping points according to their training loss rather than the number of GD iterations.

One question that arises is: do the accuracies show similar trends to the loss? Because loss and accuracy are only correlated, it's possible that even though non-linear models change loss more on easier examples, but the accuracy per-example is similar. This is somewhat answered in Figure 3 which does accuracy comparisons. However, in this scenario both the low accuracy (early time) and high accuracy (late time) are very similar across the models - making it harder to interpret the intermediate accuracy behavior.

The analysis of the theoretical model was very clean. However, it is unclear if the mechanism at play in the theoretical model is relevant for the phenomenology in real networks; the theoretical model only really allows for adjustment of the magnitude of features, not their directions.

---

> ### Author Response · Authors · 2022-11-08
> **Thank you for your comments and suggestions.**
>
> We thank the reviewer for taking the time to review our work, and for their constructive comments which helped us improve the manuscript.
>
> The reference [1] that you provided is very relevant to this paper, we added it to the related work section. In our paper, we make sure that the training dynamics stay in the linear regime by empirically computing 3 different metrics, but our choice of learning rate/scaling alpha could be systematically made using the insights from [1].
>
> We now clarify some points and describe the modifications to our manuscript as a result.
>
> *it would also be informative to see plots in terms of classification accuracy*
>
> We have added accuracy plots in appendix F (figure 11), which correspond to the experiments of figure 2. We observe similar trends, where the hierarchy between learning easy and difficult examples is more pronounced in the non-linear training regime.
>
> *accuracies as a function of c-score for the same number of GD steps*
>
> The problem with such a plot with GD steps on the x-axis (such as Figure 1.b) is that the non-linear training regime trains much faster w.r.t. number of GD steps. We discuss this in section 3.1. This is the reason for introducing our plots with the mean training loss on the x-axis. For instance, in order to reach the same training loss in the CIFAR10 experiment, we had to train for 5000 epochs in the linear regime vs 100 epochs in the non-linear training regime.
>
> *Figure 2, top left, is currently hard to parse*
>
> Following the reviewer’s suggestion, as a complement of figure 2, top left, we have added a plot in appendix F showing the evolution of the training loss differences between the two regimes.
>
> Note that another (perhaps more intuitive) way to look at Fig 2 is to compare the spread of the different loss curves which is wider in the non-linear regime (solid curves), which illustrates an enhanced ranking between groups of easy vs groups of difficult examples.
>
> *experiment on a dataset-model pair where the non-linear model performs significantly better than the linearized model*
>
> This can e.g. be observed in the experiment with label noise (and this is discussed at the end of section 3.2.2). The best test loss reached during training (if we used early stopping) is significantly lower for the non-linear model.
>
> *why [does] the gap in the solid curves [imply] that easier examples [are] being learned first*
>
> For a given clean example training loss (on the x-axis), the solid curves show the corresponding training loss on the noisy examples. While training progresses, clean examples are learned, which diminishes their training loss (from left to right). At any point, the solid curves are an indication of how much of the training loss corresponding to noisy examples remains to be learned. On the figure, we can observe that at equal clean training loss, more noisy training loss has been learned by the linear regime. In a first phase, the noisy loss increases because the model starts to make correct predictions for these examples, but their label in the training set has been flipped. In a second phase these examples are memorized with their corresponding (wrong) label, which makes the noisy loss decrease.
>
>  We clarified this point in the legend of figure 2.
>
> Reference
> [1] Temperature check: theory and practice for training models with softmax-cross-entropy losses

---

> > ### Comment · Reviewer_5RX7 · 2022-11-12
> > **Thanks for the responses**
> >
> > I believe you have answered most of my concerns. However, figure 2 top left in the main text is still unparsable to my eye. I would request that the authors try to improve this before any accepted version could be published. One possibility would be to divide into a smaller number of bins (say, 5); another could be to use a legend which gives a distinct color for each bin, rather than a colormap. I understand why the authors used a colormap and in most cases this would be appropriate; however, since this is the first figure that presents results on real models, I think it should be as unambiguous as possible in its presentation.

---

> > > ### Comment · Reviewer_5RX7 · 2022-11-12
> > > **Small edits**
> > >
> > > When referencing figures, there should be consistent notation; currently it oscillates between "Figure", "fig." and "Fig.". For Figures 4 and 14, the meaning of the colors on the left hand side plot is unclear. The MLP should be referred to as a "2-layer" MLP, not "2-layers".

---

> > > ### Author Response · Authors · 2022-11-14
> > > **more minimalistic and readable figure 2**
> > >
> > > Thank you for your suggestion. In our most recent revision, we improved figure 2 by only plotting a few bins (top 10% c-scores, bottom 10%, and 2 bins in between). We hope that this makes it easier to parse.

---

> > > > ### Comment · Reviewer_5RX7 · 2022-11-14
> > > > **Figure 2 looks great!**
> > > >
> > > > Much more readable now, thanks for the edit.

---

### Review · Reviewer_cZb7 · 2022-10-28

**Summary Of Contributions:**

This submission studies the "simplicity bias" of neural network training dynamics in different regimes defined by the model parameterization. Using various measures of example difficulty, the authors empirically show that the nonlinear feature learning regime has a stronger simplicity bias, meaning that easier examples are learned before difficult ones. The empirical findings are also supported by the theoretical analysis of a quadratic linear model, where the learning of different directions occurs at different time-scale (under sufficiently small initialization).

**Audience:**

Yes

**Claims And Evidence:**

Yes

**Requested Changes:**

Please address the points in the Weaknesses section.

**Strengths And Weaknesses:**

## Strength

The studied problem is interesting and relevant; the theory of two-layer neural networks is usually divided into two different settings: the lazy (kernel) regime and the feature learning (mean-field) regime. Most previous works focused on proving optimization and generalization guarantees for neural networks in the two regimes, but to my knowledge, how this different parameterization affects the implicit bias of gradient descent (measured by example difficulty) has not been thoroughly studied. The experimental results demonstrate an interesting separation in the learning behavior, which may inspire future research towards understanding the limitation of the NTK description of neural networks.

## Weakness

I have the following concerns:

1. The studied two-layer linear model is very idealized, and due to the quadratic parameterization, the sequentialization of learning is not surprising, as seen in many prior works on deep linear networks.

2. The notion of learning order has already appeared in many prior papers that are not thoroughly discussed in this submission:
- In idealized situations, gradient descent for kernel ridge regression already yields a "simplicity bias" where the complexity is defined by the polynomial degree, see [1][2].
- Similarly, the spectral bias of neural networks in the lazy (kernel) regime have been rigorously characterized in [3][4][5], from which we know that gradient descent learning favors large eigen-directions of the NTK, which often corresponds to low-complexity functions.
- For neural network in the feature learning regime, [6] identified a class of staircase functions where the learning of low- and high-degree components is "coupled". This allows the feature learning model to learn the difficult high-degree parts faster than the linearized kernel model.

3. [minor] References to related works are not complete.
- In the case of two-layer neural network, the feature learning regime is highly related to the mean-field regime. Please find the appropriate references for this.
- The learning order in linear networks have been extensively studied but citations are missing.
- Which theorem in [Frei et al. 2022] (cited on page 1) shows that neural network can "provably outperform any linear method"?

I have the following questions:
- In the experiments, how does the learning rate scaling affects the observed phenomena? It is known that large learning rate can alter the implicit bias and the feature learning dynamics [7] [8].
- For the theoretical analysis, would similar results holds for a vanilla two-layer linear network, where the different regimes are induced by changing the scale of initialization?
- Neural networks in the lazy regime should enjoy optimization benefits due to the exponential convergence in the training loss. In Figure 1b, why does the linearized model converge much slower (and apparently not reaching small error)?

[1] Ghosh et al. 2021. The three stages of learning dynamics in high-dimensional kernel methods.
[2] Xiao 2022. Eigenspace restructuring: a principle of space and frequency in neural networks.
[3] Bietti and Mairal 2019. On the inductive bias of neural tangent kernels.
[4] Cao et al. 2020. Towards understanding the spectral bias of deep learning.
[5] Nitanda and Suzuki 2020. Optimal rates for averaged stochastic gradient descent under neural tangent kernel regime.
[6] Abbe et al. 2022. The merged-staircase property: a necessary and nearly sufficient condition for SGD learning of sparse functions on two-layer neural networks.
[7] Wu et al. 2021. Direction matters: on the implicit bias of stochastic gradient descent with moderate learning rate.
[8] Ba et al. 2022. High-dimensional asymptotics of feature learning: how one gradient step improves the representation.

---

> ### Author Response · Authors · 2022-11-08
> **Thank you for your feedback.**
>
> We thank the reviewer for their thorough feedback on the related work.
>
> Although our submission does refer to prior work on both the simplicity/spectral bias and the learning dynamics of deep linear models (mainly in the second paragraph of the related work section),  we agree that these references need to be completed and clarified. We do so in the revised  version.
>
> Regarding the novelty of our work (points 1 and 2).  Our results can indeed be appreciated in light of the recent work on the spectral bias on one hand, and the series of theoretical works on deep linear networks on the other. Now:
>
> * Our main goal here is to compare the lazy and rich regimes. While the spectral bias is indeed well understood for gradient descent learning in linear (regression) models, our results show that the rich regime enhances this bias, as mentioned in the reviewer’s summary.
>
> * One of the main novelties  of our approach is to shift from a spectral perspective (‘Easy modes are learned first’) to the perspective of training examples (‘easy examples are learned first’). It thus bridges spectral bias analyses with the work on curriculum learning and example difficulty. The purpose of the concrete examples developed in Section 4.4 is to make this shift explicit  in our theoretical setting.
>
> * As we mention in the related work, our theoretical analysis reproduces some of the key technical ingredients of known analyses of deep linear networks (Saxe et al., 2014; Gidel et al., 2019). These  analyses already note the sequential learning of modes and its implicit regularization effect. There is a crucial difference to note, however – which we discuss more thoroughly in the revised version.  These prior analyses apply to the principal components of the input-output correlation matrix $X^\top Y$  in the context of a multidimensional output (multiclass classification or matrix factorization); in particular, the number of modes is bounded by the output dimension - hence reduces to one in the context of regression or binary classification setting. By contrast, our theoretical analysis applies to the $n$ components of the vector of labels Y in the left singular basis of the input matrix $X \in \mathbb{R}^{n \times d}$. It thus approaches the old problem of the relative learning speed of different modes in factorized models from a novel angle -- which allows us to frame the notion of example difficulty in this context.
>
> Other comments on related works:
>
> We do refer to mean field analyses (e.g., Savarese et al., 2019; Williams et al., 2019 )  in the introduction ; but we should indeed include citations in the related work section. We correct this in the revised version.  Thank you also for pointing out the very interesting  and indeed very related work ref [6] on the comparison between NTK and mean field regimes for learning staircase functions.
>
> We emphasize,  however, that we do *not* study the mean field regime  – which is an infinite-width regime in a suitable parametrization. We instead study the learning dynamics of a fixed  finite-width network, using alpha scaling that smoothly interpolates between the vanilla training regime -- where features ar learned -- and the linearized regime (described by the empirical, or finite-width NTK)  -- where they are not.
>
> On [Frei et al. 2022] : this reference is indeed not adequately referred to   –  thank you for catching this. This works studies the advantage of feature learning for an xor problem for which a model that is linear in input space does no better than random guessing - but this of course does not apply to a kernelized model such as a linearized network. This reference is removed in the revised version.
>
> Regarding your questions:
>
> * On learning rate scaling: indeed, we also expected possible confounding effects of the scale of the learning rate, as discussed  in the Appendix E of the original submission.  To clarify the issue, we performed some ablation analysis which compares learning curves and  linearity  metrics for a wide range of values for the learning rate (Fig 5 of the original submission).
>
> * The  parametrization of  the theoretical model is designed to make the training dynamics both solvable despite the nonlinearity, and  easily interpretable since it preserves the eigendirections of the feature matrix. We do not expect it to reflect the dynamics of an unconstrained two-layer  linear network.  It would be interesting to see whether vanilla  two-layer networks reproduce the behaviour shown in  Fig 4 though - we’re looking into this.
>
> * Although the feature learning regime does not enjoy the same theoretical convergence  guarantees as the linear regime, it indeed converges much faster in practice. For  theoretical work showing  how transitioning to a non-convex overparameterized objective can speed up optimization, see e.g. Arora et al, 2018.
>
> Arora et al (2018). On the optimization of neural networks: implicit acceleration by overparametrization. ICML 2018.

---

> > ### Author Response · Authors · 2022-11-11
> > **Vanilla 2-layers linear network**
> >
> > Dear reviewer,
> >
> > In our last revision, we have added some numerical experiments on vanilla 2-layers linear networks upon your suggestion, in figure 14 in appendix G. These confirm that we observe the same qualitative behavior as in the solvable setup of section 4, even though the training dynamics are more complex.

---

### Comment · Reviewer_93fy · 2022-11-27
**My criticism is adequately addressed.**

I thank the authors for their prompt reply. They addressed most of my concerns, so I am leaning toward acceptance. I agree with Reviewer 5RX7's official comment.

---

### Decision · Action_Editors · 2022-12-11

**Recommendation:** Accept as is

**Comment:**

The paper studies differences between lazy and feature learning regimes from the perspective of example difficulty. Authors empirically show that in the feature learning regime (as opposed to the lazy learning regime) networks learn the easier examples before difficult ones. The Authors also analyze toy models to theoretically demonstrate similar behavior occurs.

Overall reviewers were convinced of the correctness of the claims. While the paper focuses on simple model / datasets for experiments, reviewers pointed out that these are done well and results are interesting and proposes phenomena meritting further study in the community thus of interest to the TMLR readership.

After author response and discussions among reviewers, all reviewers recommended either accept or leaning accept. One reviewer raised that the submission is borderline. The AE agrees with reviewers recommendation and recommends accept as is. AE still strongly recommends authors to incorporate all the major/minor suggestions agreed upon with the reviewers.


**Audience:**

Paper is theoretical study in nature but also provides interesting general phenomena of deep neural networks in the feature learning regime. In theoretical deep learning, there is increasing interest in properly characterizing feature learning regime beyond lazy or NTK regime. Also example difficulty is an interesting probe to understand feature learning as authors have shown in the paper.

Quoting Reviewer `5RX7`:
"Overall the idea and experiments will be of interest to the readership of TLMR"


**Claims And Evidence:**

After author response and reviewer discussion, all reviewers were in agreement that claims made in the paper are correct and supported by empirical and theoretical analysis in the paper.

A concern raised was the use of a two layer linear model is too idealized and may not sufficiently capture phenomena in deep learning. AE believes empirical support on top of theoretical analysis is on solid grounds for demonstrating the main phenomena.